# Early Emergence Phase of SARS-CoV-2 Delta Variant in Florida, US

**DOI:** 10.3390/v14040766

**Published:** 2022-04-06

**Authors:** Eleonora Cella, Sobur Ali, Sarah E. Schmedes, Brittany Rife Magalis, Simone Marini, Marco Salemi, Jason Blanton, Taj Azarian

**Affiliations:** 1Burnett School of Biomedical Sciences, University of Central Florida, Orlando, FL 32827, USA; eleonora.cella@ucf.edu (E.C.); mdsobur.ali@ucf.edu (S.A.); 2Bureau of Public Health Laboratories, Florida Department of Health, Jacksonville, FL 32202, USA; sarah.schmedes@flhealth.gov (S.E.S.); jason.blanton@flhealth.gov (J.B.); 3Emerging Pathogens Institute, University of Florida, Gainesville, FL 32608, USA; brittany.rife@epi.ufl.edu (B.R.M.); salemi@pathology.ufl.edu (M.S.); 4Department of Pathology, Immunology, and Laboratory Medicine, University of Florida, Gainesville, FL 32608, USA; 5Department of Epidemiology, University of Florida, Gainesville, FL 32608, USA; simone.marini@ufl.edu

**Keywords:** SARS-CoV-2, Delta, early emergence, Florida, phylogenetic analysis

## Abstract

SARS-CoV-2, the causative agent of COVID-19, emerged in late 2019. The highly contagious B.1.617.2 (Delta) variant of concern (VOC) was first identified in October 2020 in India and subsequently disseminated worldwide, later becoming the dominant lineage in the US. Understanding the local transmission dynamics of early SARS-CoV-2 introductions may inform actionable mitigation efforts during subsequent pandemic waves. Yet, despite considerable genomic analysis of SARS-CoV-2 in the US, several gaps remain. Here, we explore the early emergence of the Delta variant in Florida, US using phylogenetic analysis of representative Florida and globally sampled genomes. We find multiple independent introductions into Florida primarily from North America and Europe, with a minority originating from Asia. These introductions led to three distinct clades that demonstrated varying relative rates of transmission and possessed five distinct substitutions that were 3–21 times more prevalent in the Florida sample as compared to the global sample. Our results underscore the benefits of routine viral genomic surveillance to monitor epidemic spread and support the need for more comprehensive genomic epidemiology studies of emerging variants. In addition, we provide a model of epidemic spread of newly emerging VOCs that can inform future public health responses.

## 1. Introduction

The first cases of Severe Acute Respiratory Syndrome Coronavirus 2 (SARS-CoV-2), the causative agent of COVID-19 disease, were reported in Wuhan city, China at the end of 2019 [1]. The virus subsequently spread globally, resulting in over 260 million infections and five million deaths [2]. This global account included more than 45 million cases in the United States (US) as of 30 November 2021. Several SARS-CoV-2 variants have since emerged through continuous viral genome evolution [3]. The emergence of novel SARS-CoV-2 variants of concern (VOCs) with increased infectivity, transmissibility, and/or virulence potential as compared to their progenitor lineage poses serious public health concerns. In particular, it is difficult to predict how VOCs will affect pandemic dynamics in the context of varying population immunity. Most concerning are VOCs that evade vaccine-induced immunity [4].

Emergence of VOCs in the US began in the fall of 2020 [5,6]. The SARS-CoV-2 Delta variant (B.1.617.2) was first identified in India in October 2020 and became the dominant variant causing a wave of infections from April to May of 2021. After April 2021, the variant had spread globally, impacting 65 countries across six continents [7] and was designated as a VOC by the US Centers for Disease Control and Prevention (CDC) [8] and World Health Organization (WHO) [9] on May 2021. In the United Kingdom (UK), Delta rapidly displaced the previously predominant Alpha variant (B.1.1.7), and by June 2021 was responsible for 90% of incident COVID-19 cases [10,11]. In the US, the Delta variant was first detected by mid-March 2021, and rapidly led to a new wave of infections. A nationwide study in the US reported a drop in Alpha variant cases from 70% in late April to 42% in mid-June, with the B.1.617.2 variant driving the shift [12]. In the early summer of 2021, Delta became the predominant variant in North America, as well as in South Africa, India, the UK and Canada [11,12,13,14,15].

With the emergence of the Delta variant, there was significant interest in how viral genomic diversity was associated with clinical or epidemiological characteristics. Compared to its Alpha variant predecessor, Delta possesses 12 notable substitutions in the genomic region encoding the spike protein that are associated with a fitness advantage. In particular, the D614G substitution has been associated with increased transmissibility, infectivity, and severity, relative to the original circulating strains [14,16,17,18,19,20]. This substitution is also shared with other variants including Alpha, Beta, Gamma, and more recently, Omicron. Studies have since shown that these mutations are associated with reduced neutralization of B.1.617.2 by vaccine/convalescent sera [21,22], decreased COVID-19 vaccine efficacy [19,23,24], higher viral replication efficiency [25], and an increased attack rate among younger, unvaccinated age groups [26]. For example, the prevalence of Delta variants among unvaccinated individuals was three times that among fully vaccinated individuals in the UK [26], and a recent study in South Korea indicated that pre-symptomatic transmission, superspreading potential, and higher transmissibility contributed to the high observed secondary infection rate [27].

Florida is the third-most populous state in the US, a popular tourist destination, and a conduit to the rest of the country through its 26 major airports and eight major ports. Historically, Florida has been the epicenter of notable public health events and responses such as the emergence of Zika Virus in North America in 2016 [28]. As such, the dynamics of introduction, spread, and exportation are important for the region. These dynamics are of particular importance since the emergence and subsequent spread of Delta varied considerably across geographic locations due to differences in host demographic structure, circulating variants, historical incidence, and vaccination rates [29,30,31]. At the time Delta was introduced to Florida, there were no enforced statewide travel restrictions or mask requirements, schools were conducted in person, and vaccination was limited to individuals over 40 years of age or in target risk groups. By analyzing SARS-CoV-2 viral genome sequence data from Florida in the context of global data, we sought to investigate the early emergence of Delta in Florida during a period with high background circulation of another resident variant (Alpha). Our findings provide insight into the introduction dynamics of future SARS-CoV-2 variants.

## 2. Materials and Methods

### 2.1. Dataset

All available SARS-CoV-2 genomes assigned to the Delta lineage were downloaded, excluding the low coverage and incomplete records, from GISAID (www.gisaid.org, accessed on 13 July 2021, Appendix A). Subsampling was performed to reduce the dataset for efficient computational analysis. A subsampling strategy was chosen to minimize the unbalanced representation of location-specific data while maximizing the genetic diversity and temporal distribution of sampling times for each geographic location using the Temporal And diveRsity Distribution Sampler for Phylogenetics (TARDiS) [32]. Each country was considered a distinct geographic location, with Florida considered distinct from the remainder of the United States. Subsampling 213 genomic sequences from each of these 18 locations resulted in 5500 total sequences.

Viral genome sequences were aligned using ViralMSA with default parameters [33] using Wuhan-1 (MN908947.3) as reference. Manual curation using Aliview was performed to specify start site and remove artifacts at the terminal regions [34]. Maximum-likelihood (ML) tree reconstruction with IQTREE (version 1.6.10) was employed, using the best-fit model of nucleotide substitution according to the Bayesian Information Criterion (BIC) as indicated by the Model Finder function (GTR + G + I) [35]. The statistical robustness of individual nodes was determined using 1000 bootstrap replicates. The final dataset comprised 5500 complete or near-complete SARS-CoV-2 genome sequences.

### 2.2. Phylodynamic Analysis

We sought to characterize the viral population structure and transmission dynamics through phylodynamic analysis. To this end, we estimated the change in virus effective population size (*Ne*) over time in the context of case incidence and vaccination. Estimates of *Ne* can be used to infer the growth rate of viral lineages. First, we used the ML tree to assess temporal signal, i.e., whether sufficient evolutionary change has accumulated to perform a robust phylodynamic analysis. We regressed the genetic distance of each tip (taxon) from the best-fitting root of the tree (rooted on the branch that minimizes the mean square of the residuals) with its respective sampling time using TempEst v1.5.3 [36]. Temporal signal was evidenced by a significant linear relationship, which allowed dating of internal nodes using the *treedater* package in R v3.6.0 [37,38]. To generate a time-scaled phylogeny, we used TimeTree with a constant evolutionary rate of 8.0 × 10^−4^ nucleotide substitutions per site per year to re-scale the branch lengths of the ML tree [13,39,40]. We then used TimeTree to estimate the number and source of viral introductions into Florida. For this analysis, we fit a migration model for which geographical locations of taxon sampling were assigned to external (known) and internal nodes (inferred) [40]. Using the resulting annotated tree topology, we identified putative importations and exportations by quantifying transitions between internal nodes assigned to Florida and the other geographic locations (countries and continents) included in the dataset.

The population structure of Florida Delta genomes was assessed to determine dominant sub-lineages indicative of distinct epidemic foci. Focused Bayesian molecular clock analysis was then conducted on the resulting individual clades (clade I–III) using BEAST 1.10.4 [41]. Simultaneous molecular clock calibration and coalescent *Ne* estimation was modeled using a strict molecular clock (constant evolutionary rate), HKY model of nucleotide substitution [42], and exponential growth in population size [43]. We computed Markov chain Monte Carlo (MCMC) triplicate runs of 100 million states each, sampling every 10,000 steps for each dataset. Tracer v.1.7.1 was used to evaluate effective sampling sizes for relevant parameters, using 200 as the minimum threshold for sufficient MCMC sampling [44]. Maximum clade credibility trees were summarized from the MCMC samples using TreeAnnotator after discarding 10% as burn-in.

Last, we assessed variation in relative transmission dynamics among individual clades as well as local Florida sub-trees within each clade in the context of mutational patterns. For each cluster, we calculated the *Oster* statistic, which is a function of the size in tips, the sum of the branch lengths, and the length of the longest branch within an individual subtree [45,46]. This method was recently developed to infer transmission events among persons living with human immunodeficiency virus (HIV). Though, while it uses a data-driven approach reliant on branch length information within the phylogeny, the uncertainty surrounding the application of this approach to SARS-CoV-2 transmission limits the resulting values to use in relative terms. This analysis was performed using R (code furnished upon request) [38].

## 3. Results

By July 2021, 2.4 million COVID-19 cases were reported in Florida and an estimated 11.3 million people had received at least one dose of the vaccine. A significant reduction in the number of infections in Florida was observed at the beginning of 2021, during a period that was dominated by the Alpha variant (B.1.1.7, Figure 1A). After a period of seemingly stable epidemic recession, with few new cases detected between February and June 2021, a new epidemic wave impacted the state, resulting in a significant increase in incidence. This wave coincided with the introduction of the Delta variant to the state, marked by an increase from 28.7% to 54.3% in the proportion of genomes submitted to GISAID assigned to the Delta lineage (13 July 2021, Figure 1A). Within 3 months of its introduction, the Delta variant had completely supplanted Alpha as the most prevalent variant. When the Delta wave began in March 2021, vaccination was limited to healthcare professionals, individuals >40 years of age, and persons at high risk for severe disease (e.g., immune-compromised). As a result, only 28% of the Florida population had received one vaccine dose and ~10% were fully vaccinated (Figure 1B). On 5 April, vaccine eligibility was extended to individuals >18 years of age [47] and the Food and Drug Administration approved the Pfizer COVID-19 vaccine for individuals 12 years of age or older on 12 May (Figure 1) [48].

Analysis of 5500 publicly available genomes (446 Florida genomes with a representative subset of 5054 genomes sampled globally) showed that Florida samples possessed characteristic Delta substitutions in the region encoding the spike protein with the exception of a E156G substitution that was not found among early Florida genomes (Figure 2). Five, less common substitutions emerged among Delta variants from Florida, including K77T, A222V, V289I, and V1264L, which were 3–21 times more prevalent in the Florida sample as compared to the global sample. In addition, T95I, common also among Iota and Mu variants, was found to be at a lower prevalence than the global sample (8.5% vs. 22.1%).

Linear regression of root-to-tip genetic distances against sampling dates indicated that the SARS-CoV-2 sequences evolve in a clock-like manner (r = 0.30) (Appendix A). Ancestral location reconstruction of the time-scaled tree elucidated the number and timing of viral migrations between Florida and the rest of the world (Figure 3). A total of 88 transitions from internal nodes assigned to allopatric locations to those assigned to Florida were observed in the phylogeny. Multiple inferred introductions to Florida occurred in late March to early April 2021 from India, followed by onward migration to the US starting in May (Figure 3A). Subsequent inferred introductions from other geographical locations were observed during the end of June 2021 as the Delta epidemic expanded. Overall, the greatest number of introduction events to Florida were observed to originate from the US, followed by Europe (lines marked in black, Figure 3B). Taxa from Florida were also intermediate in transitions between allopatric locations and the remaining US, accounting for nearly two-thirds of the number of migration events (double the number of inferred introductions into Florida, Figure 3B).

Independent concurrent migrations of Delta to Florida from North America, Europe, and Asia, occurring as early as April 2021, can be visualized on the ML time-tree (Figure 4). Based on internal support for divergence events within the tree (bootstrap values > 0.70), three distinct clades were identified: Subset I (*n* = 1026) comprised of 188 FL genomes; Subset II (*n* = 327) comprised 101 FL genomes, and Subset III (*n* = 470) comprised 83 FL genomes (Figure 4A, clade highlighted in grey and marked as I–III). These well-supported clades represent distinct foci of the local epidemic seeded by multiple introductions.

To assess these clades in more detail, we performed a separate Bayesian coalescent analysis (Figure 4B–D). The maximum clade credibility (MCC) trees showed four subtrees comprised predominately of Florida taxa, representing local spread (Figure 4B–D); all Florida subtrees presented with similar upper estimates of the timing of introduction-April 2021 (Figure 4B–D). However, the Florida clusters in panel C and D have very low statistical support due to the low genetic diversity of genomes sampled within a short time period [49,50,51,52]. When assessing clade-defining substitutions, we identified that spike A222V and V289I were isolated to Subset I genomes and K77T associated with Subset II. Further, a subclade of Subset I possessed both A222V and V289I, while another only carried A222V.

Comparison of transmission dynamics of each three clades and their respective subtrees found that Subset II exhibited the highest relative rate of transmission, followed by Subset III, then Subset I; however, there was no clear association between the above-described mutations and relative transmission rates. Further, there was no significant difference in geographic composition of each clade when assessed in the context of transmission rates (Figure 4). Florida subtrees exhibited a broader range of estimated transmission rates (0.62 vs. 0.14 units), which were generally higher than the background subsets (Table 1), except for Florida Subtree C and its Subset II. Florida Subtree D showed evidence of the highest relative rate of transmission (2.13 units) as compared to its progenitor clade, Subtree III.

## 4. Discussion

Genomic epidemiology has been integral in understanding SARS-CoV-2 emergence and spread and for tracking the evolutionary dynamics. In view of this, we investigated the introduction of the SARS-CoV-2 Delta variant in Florida, US from the early identification and the main peak of infection that occurred in July 2021. By June 2021, the Delta variant had become the dominant lineage in the US, unseating the Alpha variant as the most prevalent lineage in Florida, US, and abroad [12,13]. Subsequently, Delta received considerable attention due to its rapid dispersal, high transmission rate, and association with vaccine failure through vaccine breakthrough cases [15].

These findings provide insight into the introduction and emergence of a novel SARS-CoV-2 variant (Delta) to a region with moderate population-level immunity and ongoing transmission of a resident variant (Alpha). The introduction of the Delta variant to Florida was observed in early and late April 2021 primarily from India, which is consistent with the previously reported origins [13]. Multiple international introductions were observed at the end of June 2021 when approximately 45% of the state’s eligible population was fully vaccinated and approximately 53% had received at least one dose of vaccine in Florida [53].

Early Florida Delta isolates were interspersed among the isolates from other countries in the phylogeny, suggesting multiple introductions with most dating to early April 2021. These introductions coalesced into three distinct clades that dominated the early period of the Delta wave and resulted in the near-complete replacement of the resident Alpha variant. A similar dynamic of multiple repeated introductions followed by rapid interstate transmission was observed in the US after introduction of the Alpha variant [54,55]. While international and domestic travel were muted at the time Delta was introduced to Florida, there were no explicit restrictions. Likewise, as Delta completely displaced Alpha, Omicron could potentially displace Delta worldwide [56]; however, several scenarios could be considered, such as a long-term co-circulation, an Omicron wave followed by resurgence of Delta, or the emergence of Delta-Omicron recombinants. The outcome of these scenarios may be affected by the combination of immune escape and intrinsic transmissibility. This will influence the endemic state of the virus, public health response, and changes to vaccination rates or other social practices. Last, as evidenced by our analysis of relative rates of transmission, there was considerable variation in transmission dynamics of dominant Florida Delta clades, independent of mutational profile or geographic composition of the progenitor lineages that seeded the local epidemic. This points to differences in host factors or public health measures that warrant further investigation.

Although the sequence data alone do not represent the true number of epidemiologically linked transmission chains, our phylogenetic findings clearly elucidate multiple introductions of a novel variant to a population with moderate levels of population immunity, a prevailing resident variant, and the absence of mandates. When interpreting our results, it is worth noting that importations and exportations were inferred from available sequence data available in GISAID. The availability of these data is influence by several factors including SARS-CoV-2 testing and genomic surveillance capacity among countries. Further, whereas increased rates for Florida subtrees likely reflect local transmission dynamics, additional case-level data are required for more resolved investigation into the relative contribution of host and pathogen factors associated with differences. Nonetheless, our results further underscore the benefits of routine pathogen genomic surveillance to monitor outbreak investigations and support the need for more comprehensive epidemiological studies of newly emerging variants. They also provide a likely scenario for the trajectory of Omicron and subsequent variants whether emerging domestically or imported, reinforcing the necessity of continued genomic surveillance efforts globally.

## 5. Conclusions

We show that the Delta variant epidemic in Florida was seeded by multiple viral introductions, which coalesced into three distinct viral sub-populations that demonstrated distinct population dynamics. Florida also served as a waypoint for viral introductions to other geographic locations. Our results underscore the need for routine and robust viral genomic surveillance for monitoring epidemic spread. As we show here, these systems require representative geographic sampling to accurately track the emergence of VOCs. Overall, we present a model of viral emergence that can inform future public health responses and aid in assessing mitigation efforts.

As vaccine administration remains low globally due to the combination of vaccine access and vaccine hesitancy [57], the continued emergence of VOCs remains likely. To overcome this major challenge to control the SARS-CoV-2 epidemic, there is a continued need for targeted viral genomic surveillance and expansion of novel surveillance tools. Two promising approaches are the use of wastewater surveillance as well as the application of air collection monitors to detect SARS-CoV-2 in the environment [58]. When deployed, these tools can be used to assess the risk of infection in a specific setting, detect the emergence of novel variants, and track viral diversity. Furthermore, by coupling these tools with expanded viral genomic surveillance of COVID-19 cases, it is possible to create a robust network for situational awareness of the evolving pandemic.

## Figures and Tables

**Figure 1 viruses-14-00766-f001:**
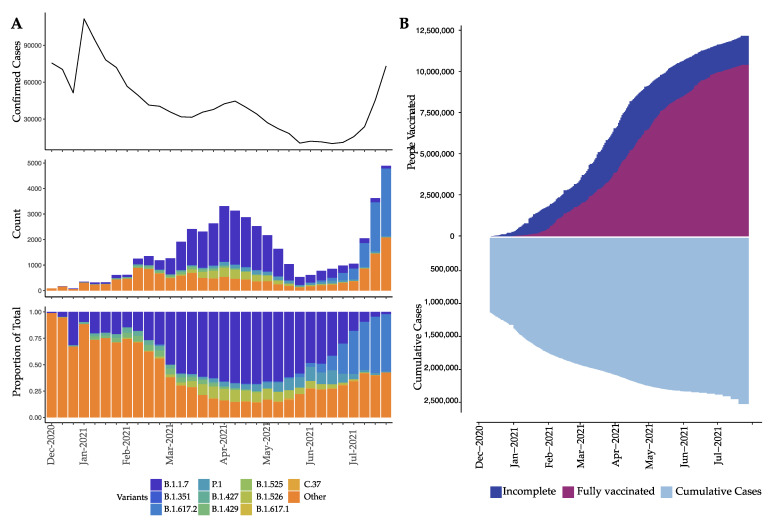
Infection distribution over time in the context of SARS-CoV-2 lineages and vaccination. (**A**) Distribution (y-axes) of previously considered variants of concern (VOCs) over time (x-axis) in terms of number of weekly confirmed cases (top), count of sequenced isolates stratified by VOC (middle), and the proportion of VOC (bottom). Remaining lineages are grouped into “Other”. (**B**) Vaccinated individuals (defined as ”incomplete” for a single dose of Moderna or Pfizer and ”fully” for two doses of Moderna or Pfizer and a single dose of Johnson&Johnson, top) and cumulative number of weekly confirmed cases (bottom) over time (x-axis).

**Figure 2 viruses-14-00766-f002:**
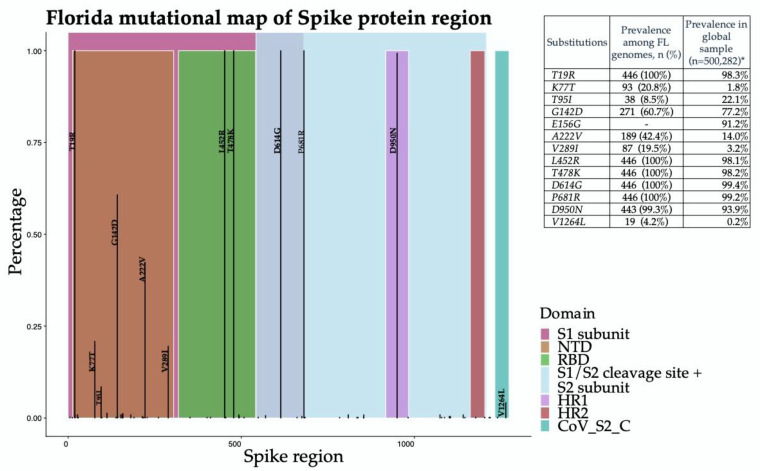
Distribution of spike protein substitutions among Florida SARS-CoV-2 Delta (B.1.617.2) variant genomes. The x-axis shows the coordinates of the region encoding the spike protein and the y-axis shows the proportion of genomes carrying the substitution. Substitutions with a prevalence greater than 2% are annotated. The inlaid table includes the Delta variant characterizing substitutions in spike protein region, the prevalence among Florida genomes, and the prevalence among worldwide Delta genomes sampled from GISAID (*) (https://outbreak.info/compare-lineages?pango=B.1.617.2&gene=S&threshold=0.2 (accessed on 13 July 2021)).

**Figure 3 viruses-14-00766-f003:**
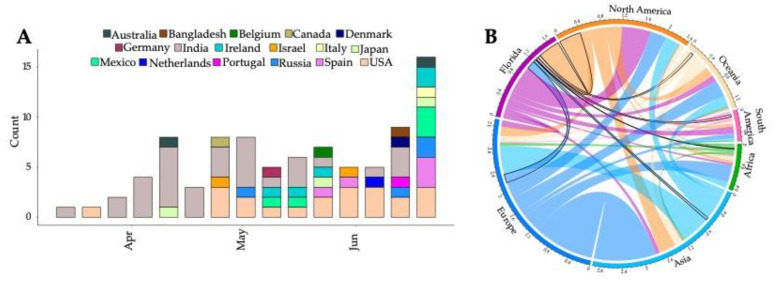
Sources of viral exchanges (imports and exports) in and outside Florida. Movement was inferred using ancestral reconstruction of geographic states within the phylogenetic tree of genomic samples. (**A**). Total number of viral introductions over time into Florida. (**B**)**.** Graphical representation of the estimated number of migration events between the geographic areas; introductions into the state of Florida are outlined in black.

**Figure 4 viruses-14-00766-f004:**
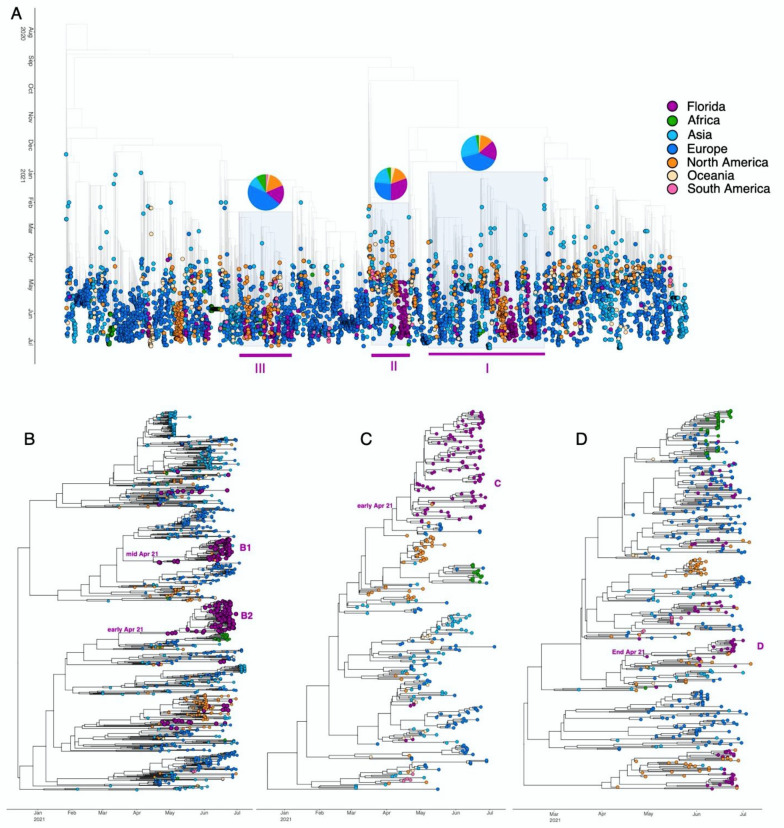
Time-resolved maximum-likelihood trees containing high-quality Delta SARS-CoV-2 near-full-genome sequences. (**A**). Full time-resolved maximum-likelihood tree for all sequence data. The three clades identified as being well-supported using bootstrap (BS) analysis (BS > 70) are highlighted in light purple (referred to as clades I–III). A pie chart indicating the location distribution is located on top of each of the three clades. (**B**–**D**). Maximum-clade credibility tree reconstruction for the three clades (I–III) individually. Tree tips are colored according to their location (continent) and the color legend is on the top right of the figure.

**Table 1 viruses-14-00766-t001:** Subset/subtree statistics (number of taxa, percentage of FL taxa, minor mutations associated and absolute Oster value).

Subset/Subtree	N of Taxa	% of FL Taxa	Minor Mutations Associated	Oster Value
Subset I	1026	18.3%	A222V, V289I	1.31
Subtree B1	74	95.9%	A222V	1.59
Subtree B2	88	100.0%	A222V, V289I	1.83
Subset II	326	30.7%	K77T	1.52
Subtree C	91	98.9%	K77T	1.51
Subset III	470	17.7%	-	1.45
Subtree D	22	81.8%	-	2.14

FL = Florida.

## Data Availability

All input files along with all resulting output files and scripts used in the present study will be made available upon request.

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
