# Peer review of "Early Emergence Phase of SARS-CoV-2 Delta Variant in Florida, US"

_viruses, 2022, doi:10.3390/v14040766_

Round 1

Reviewer 1 Report

Thanks a lot for the invitation to review this interesting manuscript.

In the current study Eleonora Cella et al described the early phase of SARS-CoV-2 B.1.617.2 lineage (the delta variant) introduction and spread in Florida, US. I believe in the value of the phylogenetics tools to conduct molecular epidemiology studies which can help to elucidate the introduction and dissemination of viruses in various settings and to help in well-informed public health mitigation efforts. 

The major results of the study showed the early multiple introductions of SARS-CoV-2 B.1.617.2 lineage into Florida mostly from North America and Europe.

Overall, the manuscript is well-written, the topic is interesrting and it fits the journal scope. The introduction provided a concise and clear overview of study topic. The authors used the state-of-the-art phylogenetic analyses tools, with enough details to replicate the results through providing the supplementary information about the GISAID sequence IDs. The results were mostly presented clearly (please check the minor points below regarding the figures). The limitations were highlighted (incomplete sampling). Thus, I suggest a minor revision for the following points:

-A few minor typographical errors were found:

Line 14: A full stop is missing before the word “Despite”.

Line 154: was extended instead of was extend

-Figure 1 can benefit from increasing the font size and using the “Palatino Linotype” font to be consistent with MDPI guidelines. The same applies for Figures 2, 3 and 4.

-In Table 1, please provide a footnote to explain the abbreviations (e.g. FL)

-In Figure 4, the statistically supported nodes (those with bootstrap values >70) were not clear; therefore, I suggest using a different color to highlight these nodes (or branches).

-Can you please provide an explanation of “Oster value” in the Methods section citing the relevant references?

-At risk of being self-serving, I suggest adding the following reference to the discussion section (lines 278-280) to highlight that high prevalence of low COVID-19 vaccine acceptance and coverage in different areas of the world (particularly in Eastern Europe, and the Middle East and North Africa), which can result in the continuous emergence of SARS-CoV-2 variants: https://www.ncbi.nlm.nih.gov/pmc/articles/PMC8760993/

-Finally, I suggest adding a short conclusions section to highlight the most significant findings of this interesting study.

Thank you!

Author Response

We thank the reviewer for their comments as they significantly improved the manuscript.  We have addressed them in detail below.   

Line 14: A full stop is missing before the word “Despite”.

Reply: We have added the full stop. Thank you for identifying this oversight.

Line 154: was extended instead of was extend

Reply: We have corrected the typo.

-Figure 1 can benefit from increasing the font size and using the “Palatino Linotype” font to be consistent with MDPI guidelines. The same applies for Figures 2, 3 and 4.

Reply:  We appreciate the suggestion and changed all of the figures accordingly.

-In Table 1, please provide a footnote to explain the abbreviations (e.g. FL)

Reply: We appreciate the suggestion, and we have included the abbreviations in the footnote.

-In Figure 4, the statistically supported nodes (those with bootstrap values >70) were not clear; therefore, I suggest using a different color to highlight these nodes (or branches).

Reply: As there are a considerable number of lineages with BS support over 70, we chose to highlight only the three clades of most interest to our analysis. Following the reviewer suggestion, we change the highlighting color.

-Can you please provide an explanation of “Oster value” in the Methods section citing the relevant references?

Reply: We expanded the methods section with a better explanation of the Oster statistic analysis.

-At risk of being self-serving, I suggest adding the following reference to the discussion section (lines 278-280) to highlight that high prevalence of low COVID-19 vaccine acceptance and coverage in different areas of the world (particularly in Eastern Europe, and the Middle East and North Africa), which can result in the continuous emergence of SARS-CoV-2 variants: https://www.ncbi.nlm.nih.gov/pmc/articles/PMC8760993/

Reply: We included this reference in the paragraph highlighting how the vaccine acceptance and coverage was low nearly worldwide.

-Finally, I suggest adding a short conclusions section to highlight the most significant findings of this interesting study.

Reply: We appreciate the suggestion, and we added a conclusion section.

Thank you!

Reviewer 2 Report

The manuscript entitled, “Early emergence phase of SARS-CoV-2 Delta variant in Florida, US,” by Cella et al. describes the phylogeography of SARS-CoV-2 in Florida, US. The major findings include identifying particular variants associated with Florida SARS-CoV-2 genome sequences as well as the identification of specific introduction events (and where they originated from) to the Florida outbreak. The results of this analysis broaden our understanding of SARS-CoV-2 outbreak dynamics.

Overall, the manuscript is well thought-out, well-written and the analyses are impressive. I have only minor suggestions to improve the flow and understanding of the manuscript:

Line 14: Missing a period after US.

Line 14: Sentence starting with “Despite” is wordy and particularly the ending should be rephrased for clarity.

Line 15: Missing a comma following elucidated

Line 31: Sentence starting with “The virus subsequently” needs a reference

Line 46: Sentence starting with ‘In the UK’ is a run-on sentence and needs to be rephrased.

Line 49: Please combine the two reference numbers into one set of brackets

Paragraph lines 55-69: Is choppy and does not flow. Please revise

Line 102: Specify which model the finder chose as the best-fit model

Paragraph 106-121: Paragraph is too wordy and does not adequately describe the methods in an understandable manner. Please revise.

Line 135: What program did you use to calculate the Oster statistic?

Paragraph lines 278-286: Paragraph is unnecessary to argue your point and should be revised or removed.

Author Response

We thank the reviewer for their detailed review.  We have addressed the comments point by point below.

Line 14: Missing a period after US.

Reply: Thank you for identifying the typo. We added the period.

Line 14: Sentence starting with “Despite” is wordy and particularly the ending should be rephrased for clarity.

Reply: We appreciate the suggestion, and we have revised the sentence.

Line 15: Missing a comma following elucidated

Reply: We addressed this typo when revising the sentence.

Line 31: Sentence starting with “The virus subsequently” needs a reference

Reply: We added the missing reference for the statement.

Line 46: Sentence starting with ‘In the UK’ is a run-on sentence and needs to be rephrased.

Reply: We appreciate the suggestion, and we have rephrased the sentence.

Line 49: Please combine the two reference numbers into one set of brackets

Reply: We merged the two citations.

Paragraph lines 55-69: Is choppy and does not flow. Please revise

Reply: We appreciate the suggestion, and we have revised the paragraph accordingly.

Line 102: Specify which model the finder chose as the best-fit model

Reply: We added in the text the best substitution model chosen according to BIC. (GTR+G+I).

Paragraph 106-121: Paragraph is too wordy and does not adequately describe the methods in an understandable manner. Please revise.

Reply: We appreciate the suggestion, and we rephrase the paragraph.

Line 135: What program did you use to calculate the Oster statistic?

Reply: The analysis was performed using R. We have included more details about the analysis in the text.  The code can be provided upon request.

Paragraph lines 278-286: Paragraph is unnecessary to argue your point and should be revised or removed.

Reply: We appreciate the suggestion. Based on reviewer 1’s comments, we have added a conclusion section and moved components of the paragraph to that.